# Influence of Cassava Morphological Traits and Environmental Conditions on Field Populations of *Bemisia tabaci*

**DOI:** 10.3390/insects12070604

**Published:** 2021-07-01

**Authors:** Kasifa Katono, Sarina Macfadyen, Christopher Abu Omongo, Thomas Lapaka Odong, John Colvin, Jeninah Karungi, Michael Hilary Otim

**Affiliations:** 1Department of Agricultural Production, School of Agricultural Sciences, College of Agricultural and Environmental Sciences, Makerere University, P.O. Box 7062 Kampala, Uganda; thomas.l.odong@gmail.com (T.L.O.); jtumutegyereize@gmail.com (J.K.); 2National Crops Resources Research Institute, P.O. Box 7084 Kampala, Uganda; chrisomongo@yahoo.com (C.A.O.); motim9405@gmail.com (M.H.O.); 3CSIRO, Clunies Ross St., Acton, ACT 2601, Australia; sarina.macfadyen@csiro.au; 4Natural Resources Institute, University of Greenwich, Chatham Maritime, Kent ME4 4TB, UK; j.colvin@greenwich.ac.uk

**Keywords:** whitefly, cassava genotype, plant morphological characteristics, population dynamics, climatic conditions

## Abstract

**Simple Summary:**

The whitefly pest complex *Bemisia tabaci* is one of the most devastating agricultural pests globally. Over the past two decades, *B. tabaci* populations reportedly increased in abundance throughout the cassava-growing regions of East and Central Africa. The current outbreak of large *B. tabaci* populations heightened its importance as a direct pest of cassava and also increased the prevalence of *B. tabaci* transmitted diseases, e.g., cassava mosaic disease and cassava brown streak disease. This undermined food security and livelihood benefits of cassava, a key staple crop in Africa with great resilience to climate change. Reasons for *B. tabaci* population increases are not well understood. Therefore, no practical options were developed for smallholder farmers in the region. To aid the design of effective and sustainable management strategies of the pest, we conducted field trials using five cassava genotypes in 2016 and 2017 to understand how the environment, cassava genotypes, and their interactions affect *B. tabaci* field populations. Our results revealed that mean monthly temperatures of 28–30 °C, total monthly rainfall of 30–150 mm, mean monthly RH of 55–70%, and development of cassava genotypes of low plant height and large leaf area and lobe width had a positive effect on *B. tabaci* population growth.

**Abstract:**

High populations of species in the whitefly complex *Bemisia tabaci* Gennadius (Hemiptera: Aleyrodidae) were reported to cause severe damage to cassava in East and Central Africa. However, reasons for *B. tabaci* population increases are not well understood. We investigated the effect of cassava morphological traits, temperature, rainfall and relative humidity (RH) on the abundance of *B. tabaci*. Five cassava genotypes with varying levels of resistance to cassava mosaic disease, cassava brown streak disease, and *B. tabaci* infestation were planted in three Ugandan agro-ecological zones. The experiment was conducted in 2016 and 2017 in a randomized complete block design. Across all locations, the tallest genotype Alado alado supported the lowest number of *B. tabaci* adults. In areas with high *B. tabaci* prevalence, leaf area, leaf lobe width, and leaf lobe number exhibited significant positive effects (*p* < 0.001) on *B. tabaci* adult count. Positive effects of relative humidity and negative effects of temperature and rainfall on *B. tabaci* adult and nymph counts were observed in 2016 and 2017, resulting in low populations in Lira. Evidently, temperatures of 28–30 °C, rainfall of 30–150 mm and RH of 55–70%, and deployment of cassava genotypes of low plant height, large leaf area, and lobe width significantly enhanced *B. tabaci* population growth.

## 1. Introduction

*Bemisia tabaci* (Hemiptera: Aleyrodidae) is among the most devastating insect pests of agricultural crops worldwide [1]. *B. tabaci* comprises a complex of 36 morphologically indistinguishable species that have a wide host plant range [2,3], including staple food crops such as cassava (*Manihot esculenta* Crantz). *B. tabaci* species complex causes three types of damage: direct feeding damage through ingestion of phloem sap, indirect feeding damage through the excretion of honeydew which results in the development of black sooty mold on leaf surfaces [4], and transmission of more than 200 plant viruses [5,6]. Outbreaks of species in this pest complex are predominantly controlled by the use of insecticides [7], which can cause further problems through disrupting natural enemies, such as predators and parasitoids that can then lead to secondary pest outbreaks. Researchers in some countries have released natural enemies to try to control outbreaks, e.g., the release of *Eretmocerus hayati* Zolnerowich & Rose in Australia [8], as well as understanding how host plant type and landscape resources support high population growth. Host plant resistance to another whitefly species, *Aleurotrachelus socialis* Bondar was successfully bred into cassava varieties in South America [9]. For non-high value crops such as cassava, especially in countries where pesticides are inaccessible to smallholder farmers, alternative control options such as host plant tolerance or resistance and natural enemies become critically important.

*B. tabaci* species complex in Sub-Saharan Africa transmit viruses that cause two economically important diseases of cassava: cassava mosaic disease (CMD) and cassava brown streak disease (CBSD) [10,11]. In Uganda, nearly 80% of the population, especially those in rural areas depend on cassava as a key food crop [12]. Limited whitefly management options are available to smallholder farmers in Uganda, and pesticides are not readily accessible. However, there is a long history of breeding cassava genotypes with desirable traits for farmers, such as resistance to disease and other insect pests such as cassava mealybug (*Phenacoccus manihoti* Matile–Ferrero) [13,14]. There is a large diversity of cassava genotypes, with very diverse form and growth habits, planted nearby each other in Uganda. This creates an ideal situation for understanding how the environment, host plant traits and cassava genotypes, and their interactions affect *B. tabaci* field populations. Identification of cassava genotype traits that consistently support low *B. tabaci* populations might be useful information for breeding programs around the world, particularly in developing countries.

As with many pests, we can say that the traits of the host plant itself interact with seasonal conditions to influence the population dynamics of *B. tabaci* [15]. However, when we compare studies from different crop types, there are conflicting results. A negative effect of temperature and a positive effect of relative humidity (RH) were observed on *B. tabaci* population size on tomato (*Solanum esculentum* Mill.) [16] and brinjal (*Solanum melongena* L.) [17] in India. Conversely, a positive effect of temperature and a negative effect of RH was also observed on *B. tabaci* population size on tomato in India [18]. In Venezuela, rainfall was reported to negatively affect *B. tabaci* population size on tomato [19]. However, researchers concluded that rainfall had a positive effect population size of *B. tabaci* on brinjal in India [17]. Plant height and canopy closure indices had no effect on infestation levels of the whiteflies *B. tabaci* (reported as *Bemisia argentifolii*) and *Trialeurodes abutilonea* on soybean in Tift county, Georgia, North America [20]. However, plant height had a positive effect on *B. tabaci* population size on chili (*Capsicum annuum* L.) in Malaysia [21], while it had a negative relationship with cotton (*Gossypium hirsutum* L.) in Pakistan [22]. 

In Uganda, the factors affecting *B. tabaci* population dynamics on cassava are not fully understood. Previous outbreaks of CMD led to the development and deployment of several CMD-resistant genotypes with no emphasis on the vector. Unfortunately, these genotypes appear to support very high *B. tabaci* populations [23], and it is not known whether these genotypes have morphological traits that enhance *B. tabaci* development. There is an increased status of *B. tabaci* as both a direct pest of cassava and a vector of CMD and CBSD [24]. *B. tabaci* was reported in most agro-ecological zones (AEZs) of Uganda in varying densities [25,26] and reported effects of climate factors on *B. tabaci* population size are variable and so far inconclusive [27,28]. Indeed, previously high populations occurred in the Lake Victoria Crescent (LVC), which is a hot and humid environment. However, relatively high populations have recently been reported in other AEZs such as the North Eastern Savanna grassland, which is a hot and dry environment. These two AEZs are characterized by the widespread adoption of disease-resistant cassava genotypes. It is not known whether these genotypes are interacting with the environment to favor *B. tabaci* population increases in certain AEZs.

It is important to determine how the environment, cassava genotypes, and their interactions affect *B. tabaci* field populations, as this knowledge will aid the design of effective and sustainable management strategies, as well as guide the development and deployment of future cassava genotypes. The objective of this study was to determine how morphological traits of cassava genotypes and changes in temperature, rainfall, and RH affect *B. tabaci* population dynamics in three contrasting regions. 

## 2. Materials and Methods

### 2.1. Field Experimental Setup

The experiment was conducted in March 2016 and March 2017. The 2016 experiment was established in two districts of Uganda: Wakiso and Lira. Wakiso is in the Lake Victoria Crescent (LVC) at 1200 m above sea level (masl) and a bimodal rainfall pattern, with an average rainfall of 1423 mm annually. Lira is in the North-Eastern Savanna grassland at 1080 m above sea level and has unimodal rainfall pattern, with an average rainfall of 1341 mm annually. The 2017 experiment was established in the same AEZs as 2016 with an addition of Kamuli (in the Kyoga plains) at 1100 m above sea level and a bimodal rainfall pattern, with an average of 1447 mm annually. These locations were chosen based on the pattern of *B. tabaci* population pressure—Wakiso, Kamuli, and Lira historically experienced high, moderate, and low populations, respectively [25,26].

### 2.2. Cassava Genotypes Used 

Five commonly grown cassava genotypes with known response to *B. tabaci*-transmitted diseases and *B. tabaci* infestation were used (as illustrated in Table 1). Apparently healthy planting materials of each genotype were selected from CBSD/CMD-free fields based on visual inspection in areas with no or very low CBSD prevalence (Apac and Lira) [29,30] and planted at each of the three sites.

### 2.3. Experimental Layout

At all locations, the experiment was laid out in a randomized complete block design with four replicate plots. The plots measured 9 m by 9 m (10 plants by 10 plants), and thus, a total of 100 plants were grown per plot. Plant spacing was 1 m by 1 m with an alley of two-meters left between blocks. Weeds were controlled manually by hand hoeing monthly for the first five months.

### 2.4. Data Collection

In 2016 at Wakiso, data were recorded monthly during 2–8 months after planting (MAP) on 10 randomly selected plants per plot. At Lira, data were collected monthly during 4–8 MAP on 15 randomly selected plants per plot. In 2017, data were recorded during 2–6 MAP in Wakiso and 2–7 MAP in Lira and Kamuli on 15 randomly selected plants per plot for the following parameters.

However, due to logistical challenges, some sample dates in Lira and Kamuli are missing.

#### 2.4.1. *Bemisia tabaci* Counts

Numbers of *B. tabaci* adults on each genotype were counted on the underside of the top five fully-expanded leaves of the tallest shoot on 15 randomly selected plants per plot. In 2016 at Wakiso, *B. tabaci* adult counts were made on 10 randomly selected plants per plot. To obtain the total *B. tabaci* nymph count, the 15th leaf from the top (which hosts the highest number of 3rd and 4th instar nymphs) was picked from each selected plant and all leaves per plot were kept in zip-lock bags and taken to the laboratory, where 3rd and 4th instar nymphs (including parasitized nymphs) on each leaf were counted using stereomicroscopes. As in other similar studies [31,32,33], raw counts for *B. tabaci* adult and nymph numbers per plant were considered in the subsequent analysis in this study. Also, the lack of relationship between nymph count and area of the leaf used as reported by [34] further justifies the use of the raw counts per plant for the analysis. 

Samples of *B. tabaci* adults were collected from the three experimental sites for genetic analysis using mtCO1 partial gene to confirm the species present at each site (for methods and results, see Appendix A
Figure A1). Results indicated that all samples were *B. tabaci* Sub-Saharan Africa 1 (SSA1) species. 

#### 2.4.2. Plant Morphological Characteristics

Leaf lobe length (of the central leaf lobe), leaf lobe width (the widest point of the central leaf lobe), leaf width, petiole length of the fifth fully-expanded leaf were measured using a meter rule (cm). Plant height was measured using a tape measure (cm) while stem girth was measured using vernier calipers (cm). Leaf lobe number was the number of lobes of the fifth fully-expanded leaf. In 2017, leaf area (cm^2^) was measured using an LI-3100C Area Meter (LI-COR Inc., Lincoln, NE, USA) and leaf area index (LAI) measured using an AccuPAR Linear PAR/LAI Ceptometer (Decagon Devices Inc., Pullman, WA, USA) in the field. In 2016, the leaf area meter and the ceptometer were unavailable, so leaf area was calculated as a product of leaf length and leaf width, and LAI was calculated using the mathematical formula for calculating the area of a circle (πr^2^) for overall plant canopy ground area.

#### 2.4.3. Environmental Parameters

Temperature and RH were recorded using a HOBO Pro v2 logger (Onset Computer Corporation) placed on one cassava plant at the center of the experiment at the trial site in each AEZ, and rainfall data were accessed from nearby weather stations. The daily readings recorded at 15:00 h for rainfall, temperature, and RH were used to calculate the total monthly rainfall, and average monthly temperature and RH values (i.e., a month was from one sample date to the next).

### 2.5. Data Analysis

All data were analyzed using R statistical software [35]. All data were tested for normality and where skewness occurred, and data were subjected to the Box Cox procedure [36] to determine the most appropriate transformation based on the lambda value. The data were then subjected to mixed model ANOVA [37] for genotype, location, and crop age (MAP) effects with replicate plots as a random effect nested within location. Means were separated using the Least Significant Difference (LSD) test at 5% probability level [38].

Scatter plots (with lines of best fit) were used to visualize the relationship between *B. tabaci* adult count and the different morphological traits in the different locations for each year [39]. Based on the visualization of the scatter plots for all locations each year, regression analysis was performed separately for each location. Subset regression [40] based on values of Schwartz’ Bayesian Information Criterion (BIC) was used to select the best multiple regression model for each location per year. Multiple linear regression [41] was run for the effect of plant morphological characteristics on *B. tabaci* adult numbers using a multiple linear regression model:**Y_i_** = ***B*_o_** + ***B*_1_X_1_** + ***B*_2_X_2_** + ***B*_3_X_3_** + … + ***B*_k_X_k_** + **ϵ_i_**
where: **Y_i_** = i^th^ response variable, ***B*_o_** is the intercept, ***B*_1_**is the regression coefficient for factor **X_1_**, ***B*_k_** is the regression coefficient for the X_k_^th^ factor and **ϵ_i_** is the error term. 

Pearson’ s correlation analysis was performed to determine the strength and the direction of the relationship between weather parameters with *B. tabaci* adult and nymph counts [42]. Data for each location in each year with all cassava genotypes pooled together were analyzed separately. 

The combined *B. tabaci* count, morphological traits, and climate data sets were then subjected to principle component analysis (PCA) [43] to understand the interactive effect of cassava genotype and environment on *B tabaci* population size. PCA was performed separately for each year. 

## 3. Results

### 3.1. Temporal Changes in B. tabaci Adult and Nymph Population on Five Cassava Genotypes in Three Locations

All main effects (location, genotype and crop age), two-ways and three-way interactions had significant (*p* < 0.001; as illustrated in Table 2) effects on *B. tabaci* adult and nymph counts in both 2016 and 2017. In 2016 in Lira, *B. tabaci* adult populations decreased steadily to reach negligible numbers at six MAP for all genotypes except NASE 14, which recorded minimum number at seven MAP (as illustrated in Figure 1). In Wakiso, fluctuations were observed in *B. tabaci* adult populations. Numbers were relatively high at two and three MAP followed by a decline to reach minimum numbers at five MAP for all cassava genotypes. Thereafter, *B. tabaci* adult populations slightly increased to reach a second peak at seven MAP, followed by a drop in the population at eight MAP for all cassava genotypes except NASE 14 whose populations reached the highest number of 10.8 adults at eight MAP (as illustrated in Figure 1). Fluctuations were also observed for *B. tabaci* nymph population in all locations in 2016 (as illustrated in Figure 1). In 2017, *B. tabaci* adult and nymph populations in all locations followed a similar trend as the nymph populations in 2016 (as illustrated in Figure 2). 

In 2016, higher *B. tabaci* adult populations were observed in Wakiso than in Lira at all crop growth stages among all cassava genotypes (as illustrated in Figure 1). However, nymph populations were higher in Lira at four and five MAP in all cassava genotypes (as illustrated in Figure 1; Appendix A
Table A1). In 2017, the highest *B. tabaci* adult and nymph populations among different genotypes were recorded in Kamuli (as illustrated in Figure 2); mean nymph count on Alado alado at two MAP was 532.9 in Kamuli, followed by 60.3 in Wakiso and 15.9 in Lira. Consistently, high mean *B. tabaci* adult and nymph populations were observed on NASE 14 and low numbers on Alado alado, in Lira in 2016 (as illustrated in Figure 1) and across all locations in 2017 (as illustrated in Figure 2). Surprisingly, in Wakiso, Alado alado had the highest mean *B. tabaci* adult population in 2016 (as illustrated in Figure 1). 

Also, the year significantly (*p* < 0.001) affected *B. tabaci* adult and nymph populations, with lower populations observed in 2016 than in 2017 (Appendix A
Table A1); in Wakiso, mean nymph count on NAROCASS 1 at three MAP was 7.5 times lower in 2016 than in 2017 (as illustrated in Figure 1 and Figure 2).

### 3.2. Status of Cassava Plant Morphological Characteristics among Genotypes in Different Environments and Their Relationship with B. tabaci Adult Abundance

All morphological traits showed significant differences (*p* < 0.001) among cassava genotypes, location, crop age, and the two-ways and three-ways interactions (as illustrated in Table 3). The most vigorous crop at all growth points among all genotypes was in Lira in both years; for example, in 2017, LAI at four MAP ranged from 3.7–4.4 in Lira, followed by 1.2–1.9 in Kamuli and 0.4–1.1 in Wakiso among all cassava genotypes (as illustrated in Appendix A
Table A2). Among all genotypes, growth was consistently high in Alado alado and low in NASE 14 in the first six months of growth in both years (as illustrated in Appendix A
Table A2). On average, Alado alado had the tallest plants in all locations in 2016 and 2017; plant height was 145–213 cm in Alado alado compared to 85–154 cm in that of NASE 14 during 4–6 MAP in Lira in 2016 (as illustrated in Figure 3A). Also, Njule Red had consistently high lobe numbers at all growth points in both years across all locations; 8.1 lobes in Lira, 7.1 lobes in Kamuli and 6.2 lobes in Wakiso at two MAP in 2017 (as illustrated in Appendix A
Table A2).

From the subset regression, plant height was selected as an explanatory variable in the regression models for all locations in 2016 and 2017 (as illustrated in Table 4). Plant height had a significant negative effect on *B. tabaci* adult counts across all locations in both years (*p* < 0.001; as illustrated in Table 4). Indeed, in Lira where the tallest crop was observed (as illustrated in Figure 3), very low numbers of *B. tabaci* adults were recorded among all genotypes (as illustrated in Appendix A
Table A1). Additionally, Alado alado, the tallest genotype across all locations in 2016 and 2017, generally had the lowest numbers of *B. tabaci* adults in all locations compared to NASE 14, the shortest genotype, on which the highest *B. tabaci* adult numbers were recorded in both 2016 and 2017 (as illustrated in Figure 1, Figure 2 and Figure 3). In Kamuli, where the highest *B. tabaci* adult populations were recorded, leaf area, lobe number and lobe width had significant positive effects (*p* < 0.001; as illustrated in Table 4) on *B. tabaci* adult count per plant; the highest *B. tabaci* adult counts of 184.8 and 167.8 were on NAROCASS 1 and NASE 14, respectively, which had high leaf area and lobe width values (as illustrated in Appendix A
Table A2). Njule Red, had many small leaf lobes resulting in a high leaf area and had relatively high *B. tabaci* adult numbers in both years. Multiple linear regression showed that the selected morphological characteristics accounted for 29% and 7% of the variation in *B. tabaci* adult count in Lira and Wakiso, respectively, in 2016. In 2017, the variation in *B. tabaci* adult count accounted for by the selected morphological characteristics was 29% in Kamuli, 27% in Lira, and 10% in Wakiso (as illustrated in Table 4).

### 3.3. Relationship and Dynamics of B. tabaci Adult and Nymph Occurrence with Environmental Factors

Correlation analysis was focused on the strength and direction of the relationship between average monthly temperature and RH, total monthly rainfall with average monthly *B. tabaci* adult and nymph count per location because of the small data set. In 2016, strong relationships of average monthly temperature (*r* = −0.85), average monthly RH (*r* = 0.83), and total monthly rainfall (*r* = 0.63) with average monthly *B. tabaci* adult count were observed in Lira. A similar relationship was observed for average monthly *B. tabaci* nymph populations (as illustrated in Table 5). In Lira, therefore, where temperatures were high, with range 29–33.6 °C, low RH of 29–57%, and high rainfall amounts, negligible numbers of *B. tabaci* adults and nymphs were recorded; the highest counts of 1.7 and 10.5 adults and nymphs per plant, respectively, were at four MAP (as illustrated in Figure 4; Appendix A
Figure A2). In Wakiso, total monthly rainfall had a strong positive effect on *B. tabaci* adult count (*r* = 0.77). In 2017, total monthly rainfall generally had a negative effect on *B. tabaci* adult and nymph populations in all locations (as illustrated in Table 5; Figure 5; Appendix A
Figure A3) with a strong association observed with adult count in Kamuli (*r* = −0.67). Maximum *B. tabaci* adult numbers were observed in months when the least total monthly rainfall was received; at four MAP in Kamuli (36.9 mm) and Lira (111.9 mm), and three MAP in Wakiso (9.3 mm) (as illustrated in Figure 5). The effects of temperature and RH on *B. tabaci* adult and nymph numbers in 2017 were inconsistent among the different locations (as illustrated in Table 5).

### 3.4. The Interacting Effect of Environment and Cassava Genotype on B. tabaci Adult and Nymph Population Size

From the PCA biplot of the different morphological traits and climate factors, a combination of high rainfall amounts, high temperatures, and large morphological trait values have a negative effect on *B. tabaci* adult and nymph counts in both 2016 and 2017 (as illustrated in Figure 6). Thus, the lowest *B. tabaci* adult and nymph counts were observed in Lira. Relative humidity has a moderately positive effect on *B. tabaci* adult and nymph counts in both 2016 and 2017. In general, three (3) distinct *B. tabaci* population size zones are revealed; the low population area (Lira), the moderate population zone (Wakiso), and the high population zones (Kamuli).

## 4. Discussion

This study aimed to determine the effects of environmental conditions and cassava morphological traits on *B. tabaci* adult and nymph populations in the field. *B. tabaci* populations (identified as SSA1 species) were present on cassava in varying numbers in all locations in both years. The variation in *B. tabaci* adult and nymph abundance in our trials demonstrated marked differences in the response of this pest to the different cassava genotypes as well as the environmental conditions in each AEZ. The mechanisms underlying these patterns may be through a combination of direct and indirect effects. Direct influences can be observed through limiting and stimulating the activity of adults and altering survival during adverse weather conditions; indirect influences can be through the effect on host plant growth and food quality [44,45].

If we focus first on the direct effects of the environmental conditions, we observed that difference in *B. tabaci* population dynamics in the different locations may be due to differences in temperature, rainfall, or relative humidity conditions. In Lira in 2017 where the highest temperature and rainfall were observed, we recorded low *B. tabaci* numbers. High rainfall amounts is known to wash away whiteflies and other insect pests, thus causing significant mortality and population reduction [46,47,48]. The negative relationship between *B. tabaci* abundance and temperature possibly resulted from reduced adult longevity and reduced reproductive success at high temperature. [49] reported mean *B. tabaci* female adult longevity of 34.10 days at 20 °C and 16.88 days at 30 °C on cucumber. [50] reported a decline in reproductive success of *B. tabaci* with increasing temperature on collard; the highest reproductive success of over 80% was at 28 °C compared to 46% for that of 33 °C. Although the monthly average temperature was similar for Lira and Kamuli during 2–6 MAP (27.9–31.9 °C), in Lira there were many days of greater than 30 °C during 6–7 MAP. Conversely, high *B. tabaci* populations were recorded in Kamuli in 2017, which experienced moderate rainfall, especially in the early crop growth stages (176.5 mm of total monthly rainfall compared to 262.9 mm at Lira at two MAP). Based on our results, the combination of high rainfall and high temperatures suppressed the population growth of *B. tabaci* in Lira. The low *B. tabaci* populations in 2016 may be attributed to differences in environmental conditions. In Lira, the low RH (29–57.1%), high temperatures (29–33.6 °C), and high rainfall (102.7–139.9 mm during 4–6 MAP followed by a drastic decline to 0 mm at 8 MAP) greatly suppressed *B. tabaci* populations. These environmental factors all contribute to low reproductive success [51] and high mortality, therefore leading to low population abundance. However, there is an urgent need for controlled studies on the effect of temperature and RH to validate our field results. 

The environmental conditions can also lead to indirect effects through differences in cassava and morphological traits express themselves across locations. Differences in plant morphology influenced *B. tabaci* population size in our trials. High growth and vigor of cassava occurred in Lira where the highest rainfall amount and temperature were recorded, supporting results of [52] who reported that cassava growth is enhanced by high temperatures if adequate rainfall is received. Thus, the cassava crop in Lira escaped the vulnerable vegetative growth stage for *B. tabaci* infestation. Therefore, there was low *B. tabaci* populations in Lira, and this was even observed on the susceptible genotype NASE 14. However, Wakiso was largely dry in the first five months, and cassava grew poorly due to moisture stress as observed by the reduced plant height and restricted leaf area, a result consistent with reports of [53] and [54]. Thus, a stressed crop that may be nutritionally unfavorable for *B. tabaci* development resulted in moderate *B. tabaci* populations as observed in Wakiso. However, in Kamuli where moderate temperature, rainfall, and RH conditions occurred in the first five months, this resulted in moderate crop vigor and cassava plants with large succulent leaves that were preferred by *B. tabaci* adults [25,55]. Hence, the highest *B. tabaci* adult and nymph abundances were recorded in this trial.

The fluctuations in *B. tabaci* adult populations across all locations in 2017, involved low initial numbers at two MAP, followed by a rapid increase and peaks at three and four MAP, and eventual decline at five MAP (as illustrated in Figure 2). Similar seasonal patterns were also observed in other studies [21,27]. This pattern may be associated with differences in host plant quality across time, with young crops with few leaves being less preferred by *B. tabaci* adults, whereas large succulent leaves produced at later crop stages may be preferred by *B. tabaci* adults. At later growth stages, the leaves senesce and the reduced food quality may prompt *B. tabaci* adults to search for better host plants for both oviposition and feeding. Previous studies demonstrated that *B. tabaci* population dynamics are strongly influenced by crop age [21,25,56]. 

Results of this study revealed that *B. tabaci* adult populations significantly varied with cassava genotype across all locations in both years (as illustrated in Figure 1 and Figure 2). This may be due to differences in plant morphological traits among the genotypes. Indeed, significant differences were observed in plant height, which was the only trait with a consistent negative effect on *B. tabaci* adults across all locations. Hence, the consistently low and high *B. tabaci* adult counts on Alado alado and NASE 14, respectively, the tallest and shortest genotypes across all locations in both years. This supports the findings of [22,57], who reported a negative relationship of *Leucinodes orbonalis* Guenee and *B. tabaci* adult populations with plant height in aubergine and cotton, respectively. In trials where the environmental conditions were suitable for growth and development of *B. tabaci* populations, we observed that leaf area, leaf lobe width, and leaf lobe number positively influenced the *B. tabaci* numbers that could be supported on cassava. Thus, high numbers were observed on NASE 14 and NAROCASS 1 in Kamuli in 2017. This is because large leaf area and lobe width provide *B. tabaci* adults a large surface area for feeding and oviposition as well as offering suitable protection from wind and rain. These findings are in agreement with [58], who reported a reduction in *B. tabaci* adult numbers on cotton with lower leaf area. Large surface area, leaf lobe width and leaf lobe number ultimately result in dense canopy cover and so high LAI values, which provides protection from high temperatures and low RH [59]. However, since this was a one study in Kamuli, further investigations in areas of high *B. tabaci* population abundance are recommended to validate these findings.

In our trials, we used farmer preferred cassava genotypes with different levels of tolerance or resistance to major cassava diseases and *B. tabaci* infestation (as illustrated in Table 1). Generally, Alado alado, which is considered tolerant to *B. tabaci*, performed as expected in terms of consistently low *B. tabaci* adult and nymph abundance in our field trials. However, Njule Red often supported similar numbers of *B. tabaci* to NAROCASS 1, NASE 3, and NASE 14 (the susceptible or moderately tolerant genotypes). Although Njule Red had very small leaf lobes as reflected in the low leaf lobe width values across all locations in both years, it had the highest leaf lobe number and leaf lobe length, which resulted in a large leaf area and high *B. tabaci* numbers. Our findings suggest that Njule Red should not be considered tolerant to *B. tabaci* in the field context.

Our trials showed that the risk of *B. tabaci* outbreaks in cassava fields is heightened by the deployment of short, broad-leafed, cassava genotypes in areas that experience a moderately hot and humid environment, with moderate amounts of rainfall. However, we did not include all factors that may be important for *B. tabaci* population dynamics in this field trial. Natural enemies such as predators and parasitoids can cause significant mortality to *B. tabaci* populations, especially late in the season [60,61]. Additionally, our plot trials were small relative to the movement capabilities of *B. tabaci* adults. Adults could easily move between plots in each trial. Although we assume that higher numbers of adults in one plot represent a biological preference for those plants over the others in the trial, there may also were a high amount of stochastic movement. Furthermore, in real smallholder production landscapes there is a diversity of cassava and non-cassava host plants that can support *B. tabaci* populations (and their natural enemies) to some degree. This diversity of resources was not represented in our field trials but may have an impact. 

## 5. Conclusions

We showed that high temperature, high rainfall, and low RH had a negative effect on *B. tabaci* abundance on cassava. Farmers are advised to establish their cassava crop at the onset of rains to maximize the benefits of heavy early season rains on *B. tabaci* population suppression and facilitate rapid crop growth. However, in areas of severe rainfall shortage (e.g., in Wakiso), irrigation is recommended to boost crop growth and pest suppression. The difference in *B. tabaci* abundance in the different AEZs calls for location-specific *B. tabaci* control measures. At sites where environmental conditions are unsuitable for the growth and development of *B. tabaci* (e.g., Lira), any cassava genotype could be deployed provided that CBSD/CMD-free planting material is used. In sites with ideal conditions, genotypes like Alado alado, which support low populations of *B. tabaci,* are beneficial; however, Alado alado is susceptible to CBSD and CMD. Therefore, the trait of high plant height observed in Alado alado could be considered by breeders in a bid to develop cassava genotypes resistant to *B. tabaci*. Our results also showed that cassava genotypes with low plant height and large leaf area and lobe width support high *B. tabaci* populations. However, these genotypes such as NASE 14 possess high levels of resistance to CMD and CBSD, and thus, are preferred by farmers. Breeding efforts should therefore be aimed at improving these genotypes with *B. tabaci* resistance traits, such as increasing plant height and reducing leaf area to avoid cassava production losses from pests and diseases. In regions of the world with *B. tabaci* species but an absence of these diseases, focusing breeding programs (or variety selection program) on these traits may also be beneficial in terms of increasing overall host plant tolerance to *B. tabaci* and reducing the ability of production landscapes to support high populations.

## Figures and Tables

**Figure 1 insects-12-00604-f001:**
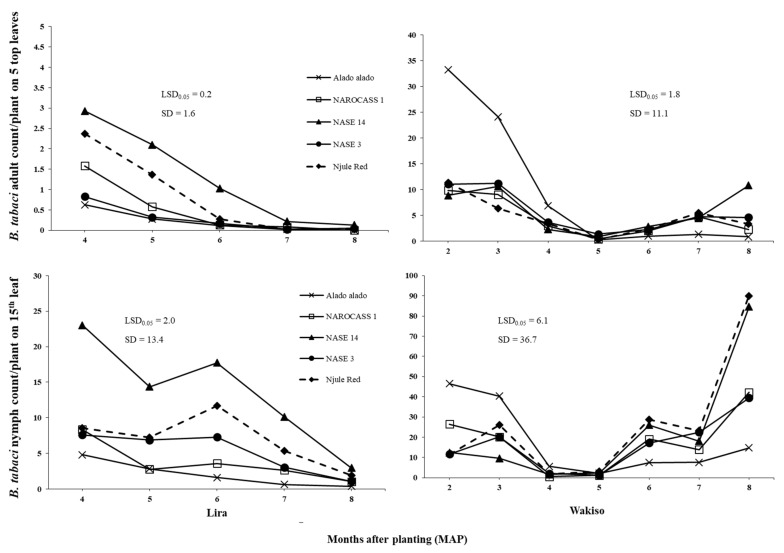
Mean monthly *B. tabaci* adult and nymph (3rd and 4th instars) counts on five cassava genotypes at different growth stages in Lira and Wakiso in 2016.

**Figure 2 insects-12-00604-f002:**
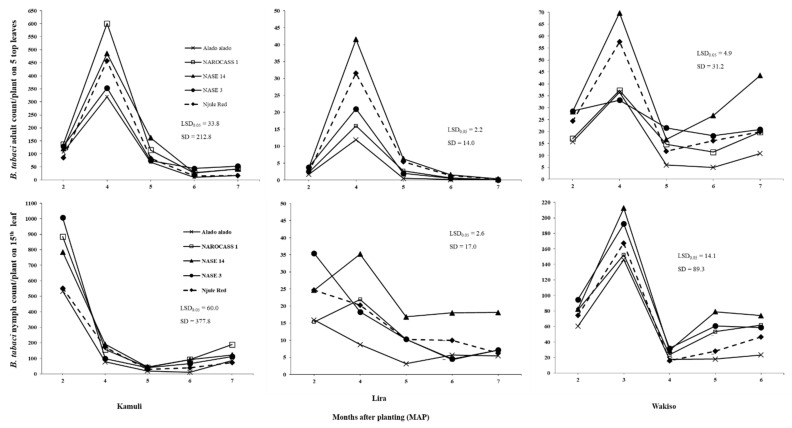
Mean monthly *B. tabaci* adult and nymph (3rd and 4th instars) counts on five cassava genotypes at different growth stages in Kamuli, Lira, and Wakiso in 2017.

**Figure 3 insects-12-00604-f003:**
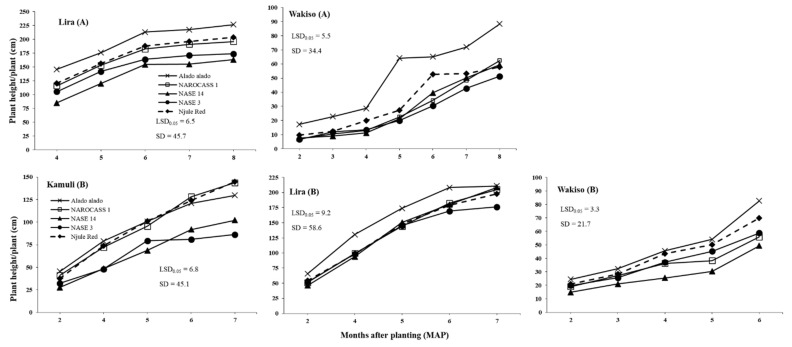
Mean monthly plant height on five cassava genotypes at different growth stages in Lira and Wakiso in 2016 (**A**) and Kamuli, Lira, and Wakiso in 2017 (**B**).

**Figure 4 insects-12-00604-f004:**
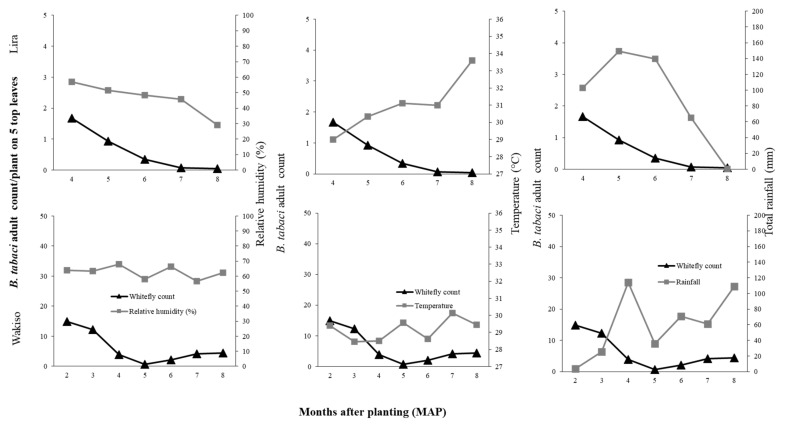
Relationships between mean monthly *B. tabaci* adult counts and monthly average relative humidity, monthly average temperature, and total monthly rainfall in 2016 in Lira (**top**) and Wakiso (**bottom**).

**Figure 5 insects-12-00604-f005:**
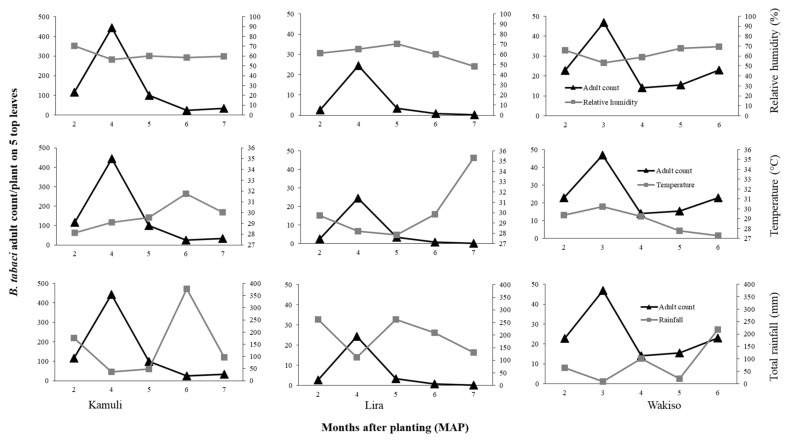
Relationships between mean monthly *B. tabaci* adult counts and monthly average relative humidity, monthly average temperature, and total monthly rainfall in 2017 in Kamuli (**left**), Lira (**center**), and Wakiso (**right**).

**Figure 6 insects-12-00604-f006:**
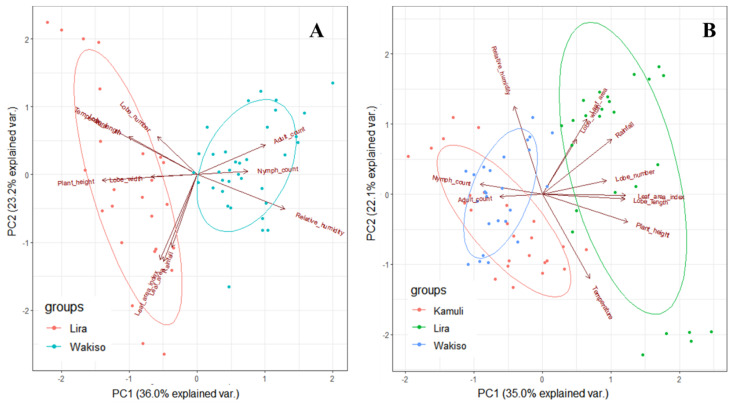
Relationship between *B. tabaci* count and cassava morphological traits and climate factors in different locations of Uganda in 2016 (**A**) and 2017 (**B**).

**Table 1 insects-12-00604-t001:** Selected cassava genotypes and their susceptibility to cassava mosaic disease (CMD), cassava brown streak disease (CBSD), and *B. tabaci*.

Genotype	Type	Reaction		
		CMD	CBSD	*B. tabaci*
Njule Red	Local	MT	S	T
Alado alado	Local	S	S	T
NAROCASS 1	Improved	R	T	S
NASE 14	Improved	R	T	MT
NASE 3	Improved	R	T	MT

S: susceptible, MT: moderately tolerant, T: tolerant and R: resistant (Omongo, C. A., personal communication, 2 March 2015; [14]).

**Table 2 insects-12-00604-t002:** F-statistics for effects of location, genotype, and crop age on *B. tabaci* adult per plant on 5 top leaves and nymph per plant on 15th leaf (3rd and 4th instars) population change over time in 2016 and 2017.

SOV	2016			2017		
	DF	*B. tabaci* Adult Count	*B. tabaci* Nymph Count	DF	*B. tabaci* Adult Count	*B. tabaci* Nymph Count
Location (A)	1	190.66 ***	66.67 ***	2	483.06 ***	109.85 ***
Genotype (B)	4	33.20 ***	70.55 ***	4	76.07 ***	133.34 ***
Crop age (C)	4	65.07 ***	101.74 ***	4	816.37 ***	591.68 ***
A × B	4	6.14 ***	7.32 ***	8	17.33 ***	14.30 ***
A × C	4	101.02 ***	290.23 ***	8	112.30 ***	101.53 ***
B × C	16	5.07 ***	7.40 ***	16	6.76 ***	11.25 ***
A × B × C	16	13.97 ***	4.92 ***	32	5.81 ***	5.48 ***
**CV (%)**		**1576.31**	**132.93**		**82.60**	**54.98**

*** = significant at *p* < 0.001.

**Table 3 insects-12-00604-t003:** F-statistics for effects of location, genotype, and crop age on cassava morphological traits over time in 2016 and 2017.

**2016**										
**SOV**	**DF**	**Plant Height**	**Leaf Area**	**LAI**	**Petiole Length**	**Leaf Width**	**Lobe Width**	**Lobe Length**	**Stem Girth**	**Lobe Number**
Location (A)	1	732.63 ***	71.86 ***	86.17 ***	116.00 ***	59.62 ***	64.85 ***	78.16 ***	534.43 ***	2.20ns
Genotype (B)	4	230.32 ***	337.82 ***	291.04 ***	251.30 ***	316.09 ***	323.08 ***	316.62 ***	13.71 ***	127.35 ***
Crop age (C)	4	624.00 ***	127.40 ***	141.18 ***	161.19 ***	136.64 ***	66.77 ***	86.73 ***	154.16 ***	21.17 ***
A × B	4	22.03 ***	9.44 ***	9.91 ***	14.94 ***	14.56 ***	61.58 ***	7.22 ***	14.96 ***	5.14 ***
A × C	4	15.29 ***	442.10 ***	512.54 ***	471.12 ***	343.44 ***	217.07 ***	416.84 ***	16.64 ***	133.12 ***
B × C	16	2.17 **	5.46 ***	3.38 ***	3.11 ***	7.24 ***	3.44 ***	3.43 ***	4.73 ***	2.67 ***
A × B × C	16	1.65 *	7.75 ***	5.77 ***	5.32 ***	6.39 ***	5.69 ***	6.00 ***	5.51 ***	3.99 ***
CV		37.49	14.35	10.76	23.29	19.49	42.98	15.68	89.89	23.09
**2017**										
**SOV**		**Plant Height**	**Leaf Area**	**LAI**	**Petiole Length**	**Leaf Width**	**Lobe Width**	**Lobe Length**	**Stem Girth**	**Lobe Number**
Location (A)	2	751.23 ***	101.92 ***	192.87 ***	140.33 ***	70.67 ***	70.67 ***	128.45 ***	214.61 ***	39.82 ***
Genotype (B)	4	245.72 ***	91.98 ***	28.05 ***	430.75 ***	290.88 ***	290.88 ***	383.69 ***	14.26 ***	284.89 ***
MAP (C)	4	2895.29 ***	218.17 ***	140.83 ***	187.70 ***	109.64 ***	109.64 ***	76.69 ***	1048.46 ***	211.90 ***
A × B	8	52.55 ***	67.28 ***	28.81 ***	47.15 ***	60.77 ***	60.77 **	54.83 ***	29.76 ***	41.55 ***
A × C	8	166.71 ***	232.12 ***	69.62 ***	159.93 ***	144.20 ***	144.20 ***	139.11 ***	24.15 ***	80.10 ***
B × C	16	5.10 ***	26.10 ***	7.01 ***	14.01 ***	27.54 ***	27.54 ***	26.64 ***	3.35 ***	9.58 ***
A × B × C	32	4.78 ***	16.06 ***	5.68 ***	8.47 ***	11.81 ***	11.81 ***	13.89 ***	2.94 ***	6.95 ***
CV		36.67	18.15	248.16	21.73	20.08	37.05	15.6	149.93	23.27

Significance: * *p* < 0.05; ** *p* < 0.01; and *** *p* < 0.001.

**Table 4 insects-12-00604-t004:** Regression between plant morphological characteristics and *B. tabaci* adult population abundance in small-plot trials from Uganda. Locations used in 2016 were Lira and Wakiso, and in 2017 were Kamuli, Lira, and Wakiso.

Morphological Characteristic	2016		2017		
	Lira	Wakiso	Kamuli	Lira	Wakiso
Intercept	3.81 ***	0.79 ***	3.42 ***	0.85 **	0.80 **
Leaf area	-	-	0.32 ***	-	0.23 ***
Plant height	−0.12 ***	−0.17 ***	−0.26 ***	−0.23 ***	−0.22 ***
Lobe number	-	0.16 ***	0.17 ***	0.26 ***	0.13 ***
Lobe length	-	-	-	-	0.19 ***
Lobe width	-	0.19 **	0.43 ***	-	-
Stem girth	-	−0.02 *	-	-	-
Petiole length	0.80 ***	-	-	-	-
LAI	1.15 ***	-	0.20 **	0.63 ***	0.19 ***
**R^2^**	**0.29 *****	**0.07 *****	**0.29 *****	**0.27 *****	**0.10 *****

Significance: * *p* < 0.05; ** *p* < 0.01; and *** *p* < 0.001; R^2^, coefficient of determination, (-) variable not selected in the regression model for a given location.

**Table 5 insects-12-00604-t005:** Correlation coefficients of *B. tabaci* adult and nymph (3rd and 4th instars) populations with RH, temperature, and rainfall in small-plot trials from Uganda. Locations used in 2016 were Lira and Wakiso, and in 2017 were Kamuli, Lira, and Wakiso.

Weather Parameter	2016				2017					
	Lira		Wakiso		Kamuli		Lira		Wakiso	
	Adult Count	Nymph Count	Adult Count	Nymph Count	Adult Count	Nymph Count	Adult Count	Nymph Count	Adult Count	Nymph Count
Temperature	−0.85	−0.92	−0.04	0.20	−0.70	−0.60	−0.74	−0.42	0.55	0.47
RH	0.83	0.97	0.25	−0.05	−0.02	0.78	0.69	0.32	−0.55	−0.40
Rain fall	0.63	0.88	0.77	0.29	−0.67	0.00	−0.19	0.01	−0.23	−0.42

## Data Availability

The data presented in this study are available on request from the corresponding author.

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
