# Peer review of "Influence of Cassava Morphological Traits and Environmental Conditions on Field Populations of Bemisia tabaci"

_insects, 2021, doi:10.3390/insects12070604_

Round 1
Reviewer 1 Report
The article "Influence of cassava morphological traits and environmental conditions on field populations of Bemisia tabaci in Uganda" presents excellent research, however, the results are applied to Ugandan regionality. As the journal Insects has a global scope, it would be important for the authors to adjust the paper to a worldwide condition. Perhaps the case of Uganda can be used as a model, which can be used in other regions that have phytosanitary problems similar to those explored in the research.
Reviewer 2 Report
This manuscript provides useful information on a major insect pest. The approach and methods appear sound. However, there are numerous issues that need addressing including: additional clarifications, awkward sentences, wordiness, repetitions, minor formatting, some overstatements/over speculations, and a few other items. The manuscript could be shortened without loss of information. Correlations were used between plant parameters and insect counts. This was appropriate. However, it seems that it would also be appropriate to test for correlation between the climate parameters and insect counts instead of regression? Throughout the manuscript, statements were made that there were cause-and- effects of either the climate or plant parameters on the whitefly populations. The experiment was not set up to confirm such effects, but was set up to test for associations. In the conclusion, it was appropriately mentioned that there may have been other factors involved. Also, there was no mentioning about if any of the plots had symptoms of whitefly-transmitted viruses or not. There was no mention of when the plants were established during each year, and if all locations were started about the same time or not. When it comes to impact of climate parameters on insect populations, means are useful, but the duration and degree of the parameters are quite important on insect biology. None of the figures included the unit of the whitefly counts. Namely, it is not clear if the counts were per plot, per plant, per leaf, per replicate, or what. To make it easier to follow the tables and figures and see how data compare among sites, it would be useful to display the same months for all sites even if no data were collected at a given site. Standard error would be useful to see with means. Figures and tables should mention that the nymph counts were for 3rd and 4th instars.
Some specific suggestions, corrections and concerns on the manuscript are as follows.
- Ln 6-12; insert a period after the last initial of each author.
- Ln 16; delete the comma after “complex”.
- Ln 16; delete “currently”.
- Ln 16 Ln 16 – spelling error; revise to: devastating.
- Ln 19; keep same tense; revise to: has heightened … and has also increased.
- Ln 20; revise “vectored” to “transmitted”.
- Ln 20; use lower case to change to “cassava mosaic disease and cassava brown streak disease”.
- Ln 21; revise “Thus, undermining” to “This has undermined”.
- Ln 22; add a period after “understood”.
- Ln 23; revise “thus” to “Therefore,”.
- Ln 23; remove the comma; revise to: “We therefore conducted”.
- Ln 27; plural; revise to: temperatures.
- Ln 27; punctuation; revise to: rainfall of 30-150 mm, relative humidity of 55-70%, and development of cassava…..
- Ln 32; revise to “However, reasons for B. tabaci ….”
- Ln 35; use lower case to change to “cassava mosaic disease, cassava brown streak disease”.
- Ln 36; wording; revise to: The experiment was conducted in 2016 and 2017 in a randomized complete block design.
- Ln 38; wording; revise to: supported the lowest number of B. tabaci adults.
- Ln 40; punctuation; revise to: Positive effects of relative humidity, and negative effects of temperature and rainfall, on B. tabaci adult and nymph counts were observed in 2017…
- Ln 42; plural; revise to: temperatures.
- Ln 45; revise “Whitefly” to “whitefly”.
- Ln 49; here and elsewhere, indent all paragraphs.
- Ln 49; revise “one of” to “among”.
- Ln 50; punctuation; revise to: worldwide [1], particularly in Africa.
- Ln 50; punctuation and wording; revise to: Bemisia tabaci currently comprises a complex of 36….
- Ln 52; insert the species name for cassava.
- Ln 54; revise “mould” to “mold”.
- Ln 56; delete “species”.
- Ln 57; use lower case to change to “cassava mosaic disease and cassava brown streak disease”.
- Ln 58; revise “thus reducing” to “This results in”.
- Ln 64; revise to “climate, and crop age”.
- Ln 67-69; wording; revise to: A negative effect of temperature and a positive effect of relative humidity were observed on B. tabaci on tomato and brinjal in India. Conversely, a positive effect of temperature and a negative effect of relative humidity was also observed on B. tabaci on tomato in India.
- Ln 68; insert the species name for both of these plants.
- Ln 71-72; revise to “…[17]. However, researchers concluded that rainfall had a positive effect on populations of B. tabaci on tomato in India [15].”
- Ln 72; revise to “The whitefly Aleurothrixus aepim (Goeldi) was reported to be negatively affected by temperature and rainfall in a study on cassava in Brazil [18].”
- Ln 73-75; revise to “Leaf area, plant height, and canopy closure indices had no effet on infestation levels of the whiteflies B. tabaci (reported as Bemisia argentifolii) and Trialeurodes abutilonea on soybean in North America [19].”
- Ln 75-77; revise to “However, plant height had a positive effect on B. tabaci on chili in Malaysia while a negative relationship was observed on cotton in Pakistan.”
- Ln 76; insert species name for chili.
- Ln 77; insert specie name for cotton.
- Ln 83; wording; revise to: It is clear that there is an ….
- Ln 85; remove comma; revise to: [24,25] and reported…
- Ln 93; wording; revise to: It is important to determine how environment, cassava genotypes, and their interactions affect B. tabaci field populations, as this knowledge will aid in the design of effective and sustainable management strategies, as well as guide the development and deployment of future cassava genotypes.
- Ln 96; wording; revise to: The objective of this study.
- Ln 101; here and elsewhere for the subsections, do not bold, use italic and insert a period after the number.
- Ln 102-106; revise to “The 2016 experiment was established in two AEZs of Uganda: Wakiso and Lira. Wakiso is in the Lake Victoria Crescent (LVC) at 1200 m above sea level and has a bi-modal rainfall pattern, with an average rainfall of 1423 mm annually. Lira is in the North Eastern Savanna grassland at 1080 m above sea level and has a uni-modal rainfall pattern, with an average rainfall of 1341 mm annually.”
- Ln 113; revise “tabaci-vectored” to “tabaci-transmitted”.
- Ln 114; of course, asymptomatic plants can be infected. Revise “Clean planting” to “Presumed clean planting”.
- Ln 118; spell out “CMD and CBSD” so that the table can stand alone.
- Ln 122-125; if the layout was the same at all locations, please mention this.
- Ln 126-131; please mention why all sample dates were not the same at all locations.
- Ln 137; were any data collected separately on parasitized nymphs?
- Ln 149; revise “Area Meter” to “area meter”.
- Ln 149; revise to “and so leaf are” to “so leaf area”.
- Ln 154; the wording is confusing.
- Ln 156; revise “i.e.” to “i.e.,”
- Ln 160; delete “in order”.
- Ln 166; delete “separately”.
- Ln 182; revise “result, all the main” to “results, all main”.
- Figure 1. Here and other figures, are the counts per plot, per plant, per leaf or what?
- Figure 1. Why are the y axes scaled to either 80 or 100 when the highest counts are less than 50?
- Ln 232; punctuation; revise to: Also, Njule Red had…
- Ln 245; Alado alado does not appear to be the tallest in 2017 in Figure 3.
- Ln 245; punctuation; revise to: 2017, generally.
- Ln 247; punctuation; revise to: genotype, on which
- Ln 250; what are the units for these counts?
- Ln 267; awkward wording.
- Ln 273; again, what are the units for these counts?
- Figure 5; what are the units for the counts? What year for these data? Standard error bars would be useful.
- Ln 294; wording; revise to: in the field
- Ln 300; revise to “respectively, supporting results of a previous study on whiteflies on cassava in Uganda [24].”
- Ln 303; revise “and RH conditions, which have” to “or relative humidity conditions, which can have”.
- Ln 305; revise “influences are through” to “influences can be through”.
- Ln 308; revise “wash away other insect pests” to “wash away whiteflies and other insect pests”.
- Ln 308-309; impact of rainfall can only physically affect insects, but can provide environments that are favorable to entomopathogens.
- Ln 310; revise “probably resulted” to “may have resulted”.
- Ln 317; the Materials and Methods section suggested that the quantity of precipitation was recorded and not the quality. Were data collected on the intensity of rainfall as is implied here?
- Ln 322; just because there may be a correlation, the data does not mean that there was a cause and effect.
- Ln 330; this is an overstatement. Significant differences do not mean “greatly influenced”; the actual data are show little or moderate differences.
- Ln 332; revise “confirming” with “supporting”.
- Ln 333; awkward wording.
- Ln 338; revise “is nutritionally” to “may be nutritionally”.
- ln 346; revise “observed by” to “observed by other studies”.
- Ln 347; revise “less attractive” to “ less preferred”.
- Ln 348; revise “are probably preferred” to “may be preferred”.
- Ln 350; revise “prompts” to “may prompt”.
- Ln 353; revise to “Results of this study”.
- Ln 353-356; this has already been said.
- Ln 356 wording; revise to: Indeed, significant differences were observed in plant height, which was the only trait with a consistent negative effect on B. tabaci adults across all locations.
- Ln 359 – wording; revise to: This supports the findings of ….
- Ln 363 – punctuation; revise to: populations, we observed
- Ln 368 wording; revise to: This supports the findings of ….
- Figs 4 and 5, and Appendix 4 and 5; the label for the X axis is “Months After planting (MAP)”; this label on the other figures is spelled “Months after planting (MAP)”; revise for consistency.
- Ln 386; entomopathogens are among natural enemies too.
- Ln 391; wording; revise to: This demonstrates a need for additional studies…
- Ln 393; wording; revise to: control measures.
- Ln 395-398; not so; the data showed that there were correlations.
- Appendix A2; insert unit for the counts; identify what “MAP” stands for. Also, the footnote says column means; clarity is needed that the significance is column per year.
- Ln 479-481; use proper journal format for the paper title.
- Ln 488; use proper journal format for the paper title.
- Ln 507-508; use proper journal format for the paper title.
- Ln 512; use proper journal format for the paper title.
- Ln 524-525; use proper journal format for the paper title.
- Ln 529-530; use proper journal format for the paper title.
- Ln 557; use proper journal format for the paper title.
- Ln 563; use proper journal format for the paper title.
- Ln 569-570; use proper journal format for the paper title.
- Ln 593-594; use proper journal format for the paper title.
- Ln 600; use italic for “Spodoptera frugiperda”.
Reviewer 3 Report
- This manuscript is well-written, although some sentences/methodologies could be clarified (such as how were the sampled plants/leaves selected, and which data set was used for correlation analysis), and some could be more precise (such as "occurrence" does not equal "abundance," and "reaction" does not equal "susceptibility" or "tolerance.")
- Although the laudable goal this study to determine the interacting influence of environment and cultivars (i.e. E x G) on whitefly abundance on cassava, the authors have largely failed to demonstrate such interaction because the influences of morphological traits/cultivars and environmental conditions (temperature, rainfall and RH) were analyzed separately. It is quite likely that whitefly abundance is a product of interaction between environment and genetics, but the authors must find a way to demonstrate that. Perhaps one way is to analyze for the interaction among plant height (the only plant morphological trait with consistent effect), environmental factors and whitefly abundance.
- I caution against making conclusion based on results from one site (out of 3 sites) and one year (out of two years), as was done on the influence of plant morphological traits on whitefly abundance in Kamuli.
- I disagree with the authors that the data had demonstrated a cyclical population dynamics. A couple of ups and downs in an eight-month period does not demonstrate consistent or cyclical pattern.
- It's not clear why the authors chose to run regression and correlation with adult abundance only.
- The authors should be cautious of whether to analyze whitefly abundance or whitefly density. I believe it is appropriate that the raw whitefly abundance (not clear from the manuscript if the pooled data from all leaves or number per leaf was used for analysis) be used for analysis to detect influence of plant morphological trait. However, whitefly density may be a more appropriate data set for analysis on the influence of environmental factors.
- The lack of equipment in 2016 for plant morphological trait measurement may have generated imprecision in the dataset. I'm glad to see that the authors analyzed the data from the two years separately.
- The major shortcomings are that the number of replication (4) was lower, plot size (81 plants) was smaller, and the number of years study was conducted (2 years) weren't as long as what similar studies were typically performed. Of particular importance to this study is that 2 years are not sufficient in gathering sufficient environmental data to clearly detect influence of environmental conditions. The authors had addressed some of these in the last paragraph. However, it is still my opinion that one more year of study would have made this a stronger manuscript.

Round 2
Reviewer 1 Report
Dear,
We will maintain our assessment of the restricted regionality of application of the results presented in the manuscript. There was no change in the text and only in the title.
Regards,
Reviewer 2 Report
The revised manuscript has been greatly improved. However, there are still a few minor items that need attention. Is it possible that the uploaded manuscript version may not be the same version as the authors had intended because the new line numbers listed in the authors’ responses mostly do not correspond with the line numbers in the pdf document? It would be useful to acknowledge in the Materials and Methods section that some sample dates were missing in Lira and Kamuli due to logistic challenges. Figures and tables should identify that the nymphal counts were for 3rd and 4th instars; each figure and table need to be able to stand alone from the rest of the manuscript.
Some specific suggestions, corrections and concerns on the manuscript are as follows.
- Ln 16; Remove the comma after “tabaci”.
- Ln 51; delete “currently”.
- Ln 23; insert a comma after “Therefore”.
- Ln 60; revise “Some countries” to “Researchers in some countries”.
- Ln 70; spell out “CBSD” when mentioned the first time.
- Ln 80-81; another term would be more suitable than “3rd world countries”; alternatives could be "low- and lower-middle-income countries”, “developing countries”, or “less developed countries”.
- Ln 91-93; the authors replied that this sentence was “Revised to ‘Plant height and canopy closure indices had no effect on infestation levels of the whiteflies B. tabaci (reported as Bemisia argentifolii) and Trialeurodes abutilonea on soybean in Tift county, Georgia–North America[19]’.” However, the text from the pdf is “Plant height and canopy closure indices had no effect on infestation levels of the whiteflies tabaci (reported as Bemisia argentifolii) on soybean in Georgia, North America [20].” Please revise to the former sentence.
- Ln 97; spell out “CMD” when mentioned the first time.
- Table 1; insert “(CMD)” and “(CBSD)” in the title of the figure so that the table can stand alone.
- Ln 368; revise “have showed” to “have shown”.
- Ln 379; insert a comma after “Thus”.
- Ln 380; awkward sentence; revise “infestation, hence the low” to “infestation. Therefore, there was low”.
- Ln 381; revise “Lira even on” to “Lira, and this was even observed on”.
- Ln 382; revise “stress hence reduce” to “stress as observed by the reduced”.
- Ln 388; insert a comma after “Hence”.
- Ln 404; insert a comma after “Hence”.
Reviewer 3 Report
The authors have made substantial improvement to the manuscript from its earlier reviewed draft. Almost all major concerns have been addressed, except for the following.
I do not think that the authors have sufficiently justified why they decided to quantify adult counts instead of adult density. In fact, I believe the authors have misunderstood what I meant. It does not matter whether a location is suitable for whitefly development or population growth or not, in the evaluation of host plant resistance, pest density is a more suitable parameter to compare (apple-to-apple) multiple accessions or cultivars, especially when the cultivars have significant difference in morphology. Is a cultivar with 120 whiteflies on a 10-sq-cm leaf (i.e. 12 whiteflies per sq-cm) really more resistant than another cultivar with 80 whiteflies on a 5-sq-cm leaf (i.e. 16 whiteflies per sq-cm)? I do not think so.
I am glad that authors agree that two years are not sufficient in generating data to assess the impacts of meteorological factors. If the authors agree on this point, I would like to see them acknowledging it somewhere in the text, justify their decision, and identify future research to address the deficiency.
It is still not clear to me how exactly did the authors corrected for the different numbers of leaves. Was the whiteflies corrected for 10 plants or 15 plants? Figure 1 does not indicate of the data presented are numbers of whiteflies per leaf or other measurement. Even if the authors decide not to reanalyze the data on per sq-cm basis, the authors should consider making it clear if the data presented are numbers of whiteflies per leaf, per 5 leaves, per 50 leaves or per 75 leaves.
If the authors cannot vouch for the precision of leaf area measurements obtained in 2016, those data should not be used.
Round 3
Reviewer 1 Report
Dear,
All suggestions were observed and inserted in detail in the text. As noted in previous reviews, the manuscript has scientific merit and deserved to receive a globalized approach. We congratulate the authors for being able to adjust the manuscript.
Regards,
Author Response
We are delighted to know that all suggestions and concerns were appropriately addressed in the previous draft.
Thank you.